# The Impact of COVID-19 on Tourist Satisfaction with B&B in Zhejiang, China: An Importance–Performance Analysis

**DOI:** 10.3390/ijerph17103747

**Published:** 2020-05-25

**Authors:** Yan Hong, Gangwei Cai, Zhoujin Mo, Weijun Gao, Lei Xu, Yuanxing Jiang, Jinming Jiang

**Affiliations:** 1School of Civil Engineering and Architecture, Zhejiang Sci–Tech University, Hangzhou 310018, China; hy@zstu.edu.cn; 2College of Civil Engineering and Architecture, Zhejiang University, Hangzhou 310058, China; xulei8563@zju.edu.cn; 3Zhejiang Province-Subordinate Architectural Design Institute, Hangzhou 310058, China; 4The Architectural Design & Research Institute of Zhejiang University Co, Ltd., Hangzhou 310014, China; 5Faculty of Environmental Engineering, University of Kitakyushu, Fukuoka 8080135, Japan; gaoweijun@me.com (W.G.); jjmwolf@outlook.com (J.J.); 6Zhejiang Tongji Vocational College of Science and Technology, Hangzhou 311231, China; jiang_yuanxing@126.com

**Keywords:** tourist satisfaction, importance–performance analysis (IPA), COVID-19, bed and breakfast (B&B), tourism resumption

## Abstract

After the outbreak of COVID-19 (especially in the stage of tourism recovery), the bed and breakfast (B&B) tourism industry faced big challenges in improving its health strategies. B&Bs are very important for the tourism industry in China and many other countries. However, few studies have studied the impact of B&Bs, under COVID-19, on tourism in China. Our paper is among one of the first studies to investigate the impact of COVID-19 on tourist satisfaction with B&Bs in China. The work/travel restrictions started from 20 January 2020, and work/after travel resumed from 20 February 2020 in Zhejiang, China. Data were collected from 588 tourists (who experienced B&Bs in Zhejiang, China) from a WeChat online survey, from 1 March to 15 March 2020. The current study attempted to fill the gap by studying the changing tourist satisfaction levels with B&Bs before/after COVID-19. Moreover, some suggestions are given to the B&B industry for tourism resumption after COVID-19 by an importance–performance analysis (IPA).

## 1. Introduction

Corona virus disease 2019 (COVID-19) is a highly infectious disease with a long incubation period [1]. It is the latest infectious disease to rapidly develop worldwide [2]. Twenty-seven cases of the unknown virus were reported on 31 December 2019 [3]. An estimated 60 million residents of Wuhan and many other cities in China were subjected to community containment measures from 23 January 2020. These large-scale types of actions have never been used in the past (even for SARS in China) [4].

One of the important goals is to minimize the economic impact of the virus on a global scale [5]. China, as the world’s most populous nation and the world’s second-largest economy, had already battled with an epidemic (SARS); at the time, however, it was 4% of the global total—it is now 17% [6]. Consumption during the first season in China will be greatly reduced: tourism (e.g., bed and breakfast (B&B)), hotels, catering, entertainment, and other traditional living service industries have suffered the most [7]. Work resumption in China was raised step by step from 20 February 2020 [8].

In 2003, a window of opportunity to modify tourism development was opened by the crisis of SARS [8]. Nature-based areas (e.g., B&Bs in the countryside) were likely to be the target destinations [9,10]. New motivations to travel to nature-based areas became evident with SARS [11]. There was a potential marketing emphasis that nature-based tourism types (e.g., nature-based B&Bs) could be invigorated and expanded after the COVID-19 crisis [12].

Thus, this article focuses on B&Bs in Zhejiang, China. There are two reasons for this: (1) the COVID-19 epidemic improved in Zhejiang. From 23 January to 1 April, 2020, there was only one death in Zhejiang. No medical staff were infected. There were also no new confirmed cases from the residents for more than 14 consecutive days. Zhejiang resumed work gradually from 20 February 2020 [13]. (2) According to a B&B market development report in 2019, the main force of the B&B was from Zhejiang [14].

The COVID-19 epidemic has been reported in many previous papers. Some researchers have reported the impact of COVID-19 on mental health in China [15,16]. However, few studies have reported the impact of the bed and breakfast (B&B) industry, under COVID-19, on tourism in China, even though it has severely affected China and the rest of the world. B&Bs were very important for the tourism industry in China and many other countries and were especially welcomed by tourists in China, United States, and other countries [17,18]. Our paper is among one of the first studies to investigate the impact of COVID-19 on tourist satisfaction with B&B in China. The time before/after satisfaction was before the work/travel restrictions (from 20 January 2020) until work/after travel resumption (after 20 February 2020). Data were collected from 588 tourists (who have experienced B&Bs in Zhejiang, China) from a WeChat online survey, lasting from 1 March to 15 March 2020. The adjusted importance (after COVID-19)–performance (before COVID-19) analysis (IPA) was used. The current study attempts to fill the research gap by investigating the changes in tourist satisfaction levels with B&Bs before/after COVID-19. Moreover, some suggestions are given to the B&B industry to recover after the COVID-19 crisis by an importance–performance analysis (IPA).

Figure 1 shows the logical model. First, this study was carried out to measure the intervening influence of B&Bs before/after COVID-19 on the correlations with tourist satisfaction levels in Zhejiang, China [19,20]. There were 588 responses that were selected (who have experienced B&B in Zhejiang, China) for the analysis. Second, descriptive statistics and an importance–performance analysis (IPA) were used to measure the impact of B&B before/after COVID-19 on tourist satisfaction levels in Zhejiang. IPA is a business research technique developed as a market tool to examine and suggest management strategies [21]. IPA prioritizes management suggestions regarding the optimal allocations that should improve tourist satisfaction. Thus, it could be a valuable practical tool for management decisions [22]. Third, some suggestions are given to the B&B industry to recover after the COVID-19 crisis by an importance–performance analysis (IPA). Moreover, suggestions of crisis preparedness and disaster–management strategies for future research are given [23]. The purpose of this article was to help the B&B industry to adapt to resumption after the COVID-19 crisis.

This article contains six sections. Section 1 is the introduction. Section 2 contains a literature review. Section 3 contains data collection and research methods. Section 4 contains the results. Section 5 comprises the impacts and limitations. Section 6 contains the conclusions.

## 2. Literature Review

### 2.1. Crisis (e.g., SARS and COVID-19) Impact on Chinese Tourism and B&B

First, natural disasters and anthropogenic environmental problems [24,25], as well as their potential to affect the image of destinations, have impacts on travel and tourism on various scales [26,27]. According to world tourism organization (WTO), in 2003, tourism arrivals fell by 1.2% to 694 million (compared to the same period in 2002) in China, and hotel occupancy rates fell by 10% [28].

Second, the number of tourists increased by 9.2% (the first two months of 2003) over the same period in 2002, and tourism revenue increased by 14.0%. After the outbreak of SARS, the number of tourists in March 2003 decreased by 6.5%, as compared to the same period in 2002. See in Figure 2 the first monthly decrease in past decades [12].

Third, how long does it take to repair the impacts of an infectious disease on tourism? The development of the crisis’ events can be divided into three periods, according to the impact on tourist flow, including the incubation period, outbreak period, and recession period. The impact time of most crisis events is within one year; the impact period of a few events was around two years. Taking SARS as an example, the peak period of impact was from March to June 2003, and the entire impact period was about 1 year [12]. Taking the accommodation industry as an example, during 2003, single-store revenue of in high-star hotels declined significantly. B&B grew by 15.2% in 2003 and continued to grow, resulting in a 22% growth in 2004. Therefore, the impact of SARS on B&Bs in tourism was basically eliminated about half a year after the end of SARS [28].

### 2.2. Tourism Resumption Post-Crisis (e.g., SARS and COVID-19)

First, the destination image is defined as an individual’s mental representation and overall perception of a particular destination [29]. Destination image and tourist satisfaction are also important tools to actively research and manage the perceptions of tourists about the destination [30,31]. The key of the before/after crisis themes that emerged included a lack of disaster-management plans, damage to destination image and reputation. It also included the changes in tourist behavior during the crisis (e.g., COVID-19) [23]. To influence the destination choice decision-making process and to condition the after-decision-making behaviors, including participation [32], satisfaction, and future intention (e.g., sustainable mountain tourism [33,34]) to revisit [35]. The destination image is generally interpreted as impressions based on information processing from various sources over time that results in a mental representation of the attributes and benefits sought in a destination [36].

Second, our focus on post-crisis recovery is required because much of the research relates to tourism crisis (e.g., COVID-19) management [37]. The recovery should be taken as more than just an industry or economic approach, and should focus on pre-event levels [38,39,40]. The importance of the relationship between marketing with tourist satisfaction and suggestions to repair destination images was identified [23].

### 2.3. The Concept of B&B

First of all, apart from hotels and guesthouses, the most common form of accommodation is bed and breakfasts (B&Bs), which is a concept that originated in Europe [41]. These refer to small hotels that provide a non-commercial, home-like environment and only serve breakfast [41]. This also means that visitors or guests pay to stay in a private residence and interact with a local family [42]. B&Bs allow tourists to seek lodging for the night, especially when hotels and inns are unavailable in remote areas [43]. Second, the basic standards are different from other types of hotels are. The differences include B&Bs being small scale, family operated and providing special services [44]. In recent years, the B&B industry has become a unique and rapidly growing industry in the hotel industry [45]. This operation attracts tourists with different standards than hotels [46].

### 2.4. B&B in Zhejiang

This study was carried out in Zhejiang, China. As the most popular B&B rural tourist destination in China, the area receives more than 23.52 million tourists. From 1 January 2015, to 14 December 2019, the Baidu index results showed that the top 10 B&B provinces and cities were Zhejiang, Guangdong, Sichuan, Jiangsu, Beijing, Shanghai, Shandong, Henan, Chongqing, and Hubei. It shows that Zhejiang was the most concerned about B&Bs. Most of these areas are economically developed provinces and cities. According to the B&B Market Development Report in 2019, the main force of the B&B was from Zhejiang Province and accounts for about 60% of tourists, which is consistent with the search results of the area where B&Bs are present [47]. The highest media coverage about B&Bs in China was in Zhejiang from January 2015 to February 2018. The topics of media concern ranged from the rapid development of B&Bs and the reference of B&B experience to the problems arising in the development of B&Bs and lasted until the introduction of B&B standards, which indicates that the development of B&Bs in China entered a stable development stage from the initial stage of rapid growth without supervision [47].

### 2.5. Tourist Satisfaction

Various definitions of satisfaction have been proposed in the literature. In the tourism sector, tourist satisfaction (TS) is an essential aspect of the tourist services sector [48]. As services directly impact people [49], some researchers have indicated that services are linked to tourist satisfaction [41]. Tourist satisfaction, as a marketing tool, plays a key role in the construction of strategies in the tourism market [50]. Furthermore, satisfaction is vital for successful destination marketing [51], as well as a service organization [52]. Feelings of pleasure by tourists are a sign of satisfaction [53], while tourists who enjoy visiting are satisfied [54,55]. Therefore, tourist contentment is a considerable factor for tourists in making up their minds to visit again or not [56,57,58].

Enhancing tourist satisfaction is a key strategy that leads to the success of companies in the hotel [59,60], catering [61,62], and tourism industries [63]. The quantitative approach with surveys was extensively adopted by scholars to study the multiple determinants of tourist satisfaction [64]. For instance, Deng et al. [65] surveyed 412 overseas tourists in Taiwan and found tourist complaints and service quality were related to tourist satisfaction. Kim et al. [66] gathered the opinions of 317 tourists from Beijing and discovered that convenience, safety, and technological inclination were the main factors that influenced tourist satisfaction. As a unique style of accommodation, it was inappropriate to employ the factors identified in other contexts directly to B&B during our investigations [67,68,69].

## 3. Materials and Methods

### 3.1. Data Collection

#### 3.1.1. Explanation of Questionnaire

This study was carried out to measure the intervening influence of B&Bs before/after COVID-19 on its relationship with tourist satisfaction in Zhejiang, China. We used WeChat (Tencent, Shenzhen, China) for this online survey in Zhejiang, China. We received 1120 answers to the questionnaire. However, there were 588 responses from people who have experienced B&Bs in Zhejiang before the COVID-19 that were selected for the analysis. The responses were collected from 1 March to 15 March 2020.

The questionnaire consisted of 30 factors, from the expectation of B&Bs before check-in, to the perception of facilities after check-in [70]. Likert’s five-point scale was used to measure tourists’ expectations before check-in, with five optional levels [71]: (1) Importance (After COVID-19): “5 = very important”, “4 = important”, “3 = so–so”, “2 = unimportant”, and “1 = very unimportant”. Appendix A shows the sample questionnaire [72,73]. (2) Performance (Before COVID-19): “5 = very good”, “4 = good”, “3 = so–so”, “2 = not good”, and “1 = bad”.

#### 3.1.2. Questionnaire Items

In addition to people’s natural awareness and sharing awareness, the development of home-stays is more about providing experiential services for tourists than those provided by basic accommodation services [74]. Some researchers constructed an experiential scale to tap into tourist experiences in the accommodation industry [75,76]. The determinants of consumer satisfaction with B&B establishments were studied and a hierarchical structure of these determinants was built. Thus, with the intention of bridging this gap, we aspired to develop a multiple-item scale to measure tourist opinions about B&Bs before/after COVID-19.

Ten determinants of tourist satisfaction were identified [77]. Based on previous research and B&B industry evaluation standards (BIES) in China (Table 1, Figure 3), a number of factors were generated. All the factors were assessed for content and face validity by a panel of experts from two institutions affiliated with the authors [78,79].

### 3.2. Importance–Performance Analysis (IPA)

#### 3.2.1. Concept of IPA

Importance–Performance Analysis (IPA) is a business research technique developed as a marketing tool to review and suggest new management strategies [21]. While it was originally developed for marketing purposes, its applications have expanded to various fields, including tourism [90,91,92], healthcare [93,94], sustainable cities [95], social and economic outcomes [96,97], etc. The main goal of IPA is to diagnose the performance of different product or service attributes while facilitating data interpretation and providing practical recommendations for management [98]. IPA can gain insight into which product or service area managers should be targeted by identifying the most critical attributes, strengths and weaknesses [99]. The IPA technique combines measures of tourists’ perceived performance and importance into a two-dimensional plot to facilitate data interpretation [21]. Thus, each quadrant in the standard IPA chart represents a different strategy that can help managers to identify areas of concern and necessary measures to increase tourist satisfaction [100]. Choosing the right attributes to measure importance and performance is essential for obtaining the best management decisions because these decisions rely on the information revealed from the selected attribute set [101]. Figure 4 shows the IPA model.

#### 3.2.2. Location of the Discriminating Thresholds within the IPA Plot

The best place to divide the graph into quadrant thresholds is one of the biggest problems in IPA applications [102]. The choice of threshold is almost a matter of judgment [103,104]. However, their subjective positions has led to inconsistencies in existing IPA research results [104].

First, the data-centric (DC) method uses the actual data average of the observed importance and performance level as the critical point. Therefore, scholars have proposed another solution [105,106,107]. That is, they set the mean of the experience gained from the data as the intersection [108].

Second, some authors suggested that the SC method is more transparent when interpreting research results and usually provides a simpler description than using actual data methods [104]. However, using the scaling method has a serious drawback, that is, in addition to the fact that it is not driven by actual data, it also tends to record the high importance level of all the attributes. The latter means that, regardless of the characteristics of the interviewees, it turns out that this is the determinant of their expectations and opinions [109]. Each survey will have the same discrimination threshold. Incorrect threshold settings may lead to misleading and conflicting management recommendations [22].

Third, other researchers used diagonal lines (DL) or so-called isolines (IRL) to divide the plot into two separate areas [103]. The point on this 45° upward line indicates an attribute with the same importance and performance level; compared to the subjective threshold selection method, the IRL method can be said to be a more suitable method for identifying the area of interest because it directly focuses on satisfaction and importance grade difference [22]. Rial et al. [110] simplified this method by empirical means and a diagonal line with discrepancies. The difference in attributes (distance from the diagonal) is considered to be a priority in improving the service [108].

Linear relationship (or linear association) is a statistical term used to describe the linear relationship between variables and constants. Mathematically speaking, the linear relationship satisfies the equation:(1)y=mx+b.

In this equation, “*x*” and “*y*” are the two variables associated with parameters “*m*” and “*b*”. Graphically, *y* = *mx* + *b* is drawn on the *x*-*y* plane with the slope “*m*” and *y*-intercept “*b*”. When *x* = 0, *y*-intercept “*b*” is just the value of “*y*”. Calculate the slope “*m*” from any two separate points (*x*1, *y*1) and (*x*2, *y*2), as follows:(2)m=(x2−x1)/(y2−y1).

However, compared to the standard IPA chart (with four quadrants), it produces less information, provides limited identification ability and therefore has limited interpretation ability. Therefore, it limits the usefulness of IPA [22].

The actual means of importance and performance are likely to differ in most cases, and therefore, require study-specific adjustments to the scales in order to interpret the importance and performance ratings [111], as well as the relative interpretation of attributes within the importance and performance ratings [22]. Most researchers use DC and average values of the actual importance and performance level when determining the threshold value of tourism research [108]. IRL directly focuses on differences in satisfaction (Before COVID-19) and importance (After COVID-19) ratings. Therefore, this article uses the method of DC+IRL when specifying the thresholds of the impact before/after COVID-19 on tourist satisfaction with B&B (Figure 5).

## 4. Results

### 4.1. The Descriptive Statistics

#### 4.1.1. Profile of Survey Respondents

Table 1 describes the respondents’ demographic profile [96,97,98]. Among the 588 tourists, 55.78% were women and 44.22% were men. The majority of the participants ranged from 25 to 35, accounting for 45.92% of the samples. Most of the respondents had a bachelor or graduate degree (51.02%, *n* = 300), followed by graduate degrees (32.31%, *n* = 190). With regards to monthly income, 45.92% (*n* = 270) reported that their annual income was between $801 and $1200.

#### 4.1.2. Reliability and Validity Analysis

The statistical software of SPSS 26 (IBM, New York, NY, USA) was used in the questionnaire analysis [112,113]. The calculation of the questionnaire’s reliability was based on the Cronbach’s Alpha coefficient [101,102]. An α larger than 0.7 indicates “highly reliable” and larger than 0.5 “reliable” [114,115,116]. The α for this questionnaire was 0.978, which indicated a relatively high and acceptable reliability [71,90]. The questionnaire also proved satisfying in terms of the content validity, criterion-related validity, and construct validity (Table 2, Table 3).

Cronbach’s alpha is a function of the number of test items and the average inter-correlation among the items. It showed the formula of the Cronbach’s alpha below [114]:(3)$$\alpha =\frac{N*\overset{ˉ}{c}}{\overset{ˉ}{v}+(N-1)*\overset{ˉ}{c}}.$$

Here, N is equal to the number of items, $\overset{ˉ}{c}$ is the average covariance between the item-pairs, and $\overset{ˉ}{v}$ is equal to the average variance. It can be seen from this formula that, if you increase the number of items, you will increase Cronbach’s alpha. In addition, if the correlation between the average items is low, the alpha will be low. As the correlation between the average items increases, Cronbach’s alpha will increase (keeping the number of items unchanged).

#### 4.1.3. Importance–Performance Scores

The mean responses for the importance and performance of the 30 attributes were analyzed in accordance with the IPA framework and are shown in Table 4. Most of the importance and performance means (Table 5.) were found to be significantly different (Sig. 2–tailed) at the <0.01 level (QN. 23/25/28/29/30 <0.05) [117]. Variables in each category were ranked in order by Paired Differences (IA–PB) [118,119].

### 4.2. Importance (after COVID-19)–Performance (before COVID-19) Analysis (IPA)

Figure 6 shows results of the analysis. High priority area (part of Quadrant 4+3): (Quadrant 4) the “concentrate here” area; (Quadrant 3) the “low priority” area. The attributes in this quadrant were considered to perform poorly and therefore represent the main weakness of the product and a threat to its competitiveness. In terms of investment, these attributes have the highest priority [98]. Rank by paired differences (IA–PB): (1)“The layout of the rooms is scattered”, (2) “Split air conditioners are used in guest rooms”, (3) “Places or items for cleaning and disinfection are provided to tourists”, (4) “Rooms of Single B&B are few and exquisite”, (5) “Contingency plans are developed and can be exercised regularly”, and (6) “Buildings are intelligent (e.g., semi–self–service management)”.

Priority area (part of Quadrant 1+3): (Quadrant 1) the “keep up the good work” area; (Quadrant 3) the “low priority” area. It represents the main and potential competitive advantages of a product or service. Attributes in this quadrant are considered to be performing well and investment needs to continue. Rank by paired differences (IA–PB): (7) “Rooms are naturally ventilated”, (8) “The emergency facilities are complete (such as: first aid kit, escape equipment)”, (9) “The rooms are spacious and clean”, (10) “The outdoor space is large and natural (e.g., courtyard, terrace, roof garden)”, and (11) “Green consumption is encouraged and environmental protection measures are implemented”.

Medium priority area (part of Quadrant 1): (Quadrant 1) the “only keep up the good work” area. The importance and performance of these factors (e.g., “location and nearby facilities are safe and good”, “other service rooms are clean and tidy”) were good.

Low priority area (part of Quadrant 2+3): (Quadrant 2) the “possible overkill” area; (Quadrant 3) the “low priority” area. Their performance was not particularly good, but they were considered relatively unimportant to tourists; therefore, managers should not pay too much attention to these attributes. They represent a slight weakness, and poor performance is not a big problem. These factors are not important in this article (e.g., “the indoor and outdoor transition spaces are natural and beautiful (e.g., gallery frames, awnings, balconies)”, “the shading performance is good (e.g., opaque curtains))”.

## 5. Discussion

### 5.1. Impacts

#### Implications

First, to the best of our knowledge, this study is among the first to uncover the impact of COVID-19 factors influencing tourists’ satisfaction with B&Bs. Second, from the perspective of methodology, DC and IRL were combined with content analysis to sort and guide the complexity of the relationship between variables, which has certain value for future research. Third, some suggestions would be given to the B&B industry to recover after COVID-19 by the importance–performance analysis (IPA). Our study extends this research area from the traditional B&B context and adds knowledge to the post–COVID-19 B&B tourism management area.

### 5.2. Suggestions

This study also has practical suggestions for B&B operators in making marketing strategies after COVID-19. As our results show, psychological factors can directly affect the satisfaction of consumers after Covid-19. The managers of B&Bs should consider the following factors in the “High priority area” and “Priority area”. Compared to before COVID-19, tourists were more concerned with the natural and safe experience associated with B&Bs after COVID-19. These are some suggestions to improve consumption experience.

#### 5.2.1. High Priority Suggestions

Suggestions for the rank by paired differences (IA–PB): (1) “The layout of the rooms is scattered”. This shows that, after the COVID-19 epidemic, tourists prefer scattered room layouts. Centralized room layouts need to be reconsidered. (2) “Split air conditioners are used in guest rooms”. At present, central air conditioning has been used by many B&Bs. However, after COVID-19, this is not an ideal choice. (3) “Places or items for cleaning and disinfection are provided to tourists”, and (4) “Rooms of Single B&B are few and exquisite”. At present, there are more and more rooms in many B&Bs (single) and the scale is getting larger. This is obviously inappropriate for B&B tourism after the COVID-19 epidemic and it needs to be changed. (5) “Contingency plans are developed and can be exercised regularly”. These measures were not paid enough attention before the outbreak. It needs to be focused on after the COVID-19 epidemic. (6) “Buildings are intelligent (e.g., semi–self–service management)”. “Intelligentization” will be a trend in the future. It also needs to be focused on in B&Bs.

#### 5.2.2. Priority Suggestions

Suggestions for the rank by paired differences (IA–PB): (7) “Rooms are naturally ventilated”, (8) “The emergency facilities are complete (such as first aid kit, escape equipment)”, (9) “The rooms are spacious and clean”, (10) “The outdoor space is large and natural (e.g., courtyard, terrace, roof garden)”, and (11) “Green consumption is encouraged and environmental protection measures are implemented”. The suggestion in this part is that more attention should be paid to nature and green areas in the B&B tourism after COVID-19. Just like after SARS in 2003, people tended to go to places with nature-based areas rather than urban vacations [7,43].

## 6. Conclusions

The priority suggestions in this paper will be of great help to improve the attraction of B&Bs to tourists after Covid-19. Enough attention was not paid to these measures before the outbreak. They need to be focused on after the COVID-19 epidemic. B&Bs are very important for the tourism industry in many countries, and tourists have especially welcomed them in recent years in China. To the best of our knowledge, our study was among the first to investigate the immediate impact of the COVID-19 pandemic on tourist satisfaction with B&Bs in China. Many previous studies have reported on COVID-19. Some others studied the correlations between COVID-19 and the quality of life in China. However, few studies have reported the impact of B&B under COVID-19 on tourism in China. The adjusted importance (after COVID-19)–performance (before COVID-19) analysis (IPA) was a new attempt. Moreover, some promotion suggestions were given to the B&B industry recovery after COVID-19 by the IPA.

However, there were some limitations to our study and future research areas. First, the data were collected from tourists in B&Bs in Zhejiang, China. Thus, it was somewhat difficult to apply the suggestions of the impact of COVID-19 to other areas. Future researchers may expand this scope. Second, although we identified the relationships between the determinants of tourist satisfaction and COVID-19, the relative strength of these correlations was unknown. We can test the model and identify the degree of influence of the correlations between these factors to promote the B&B industry in further research. More nuanced research questions should be incorporated. Third, the current paper employed an IPA approach. Even though this method is a widely known method in the tourism industry, it was also a new attempt to use the IPA model with B&Bs. Thus, we suggest that researchers in other parts of China and in other continents work together to produce similar studies, thereby creating a worldwide body of literature that examines the phenomena related to the effects of crises (e.g., COVID-19) and their impact on B&Bs and tourism.

## Figures and Tables

**Figure 1 ijerph-17-03747-f001:**
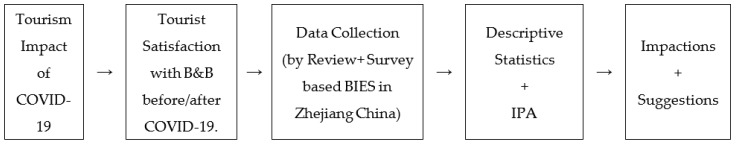
The logical model. Notes: B&B = Bed and Breakfast, BIES = B & B industry evaluation standard, COVID-19 = Corona Virus Disease 2019, IPA = importance–performance analysis, IA = Importance (After COVID-19), PB = Performance (Before COVID-19), QN = Question Number, TS = tourist satisfaction.

**Figure 2 ijerph-17-03747-f002:**
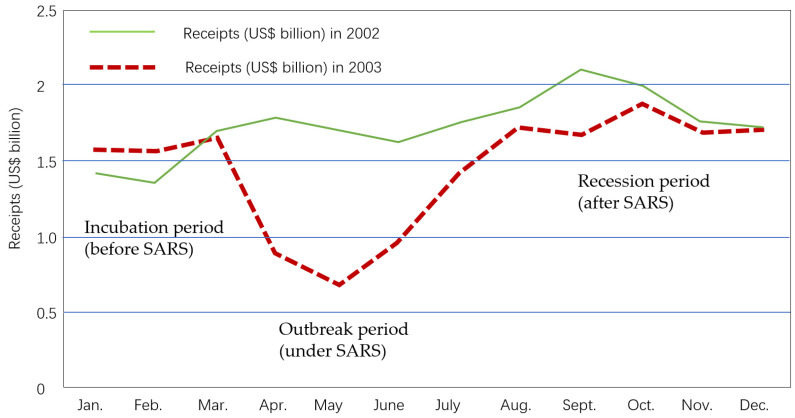
The impact of SARS on Chinese tourism between 2002 and 2003 [12].

**Figure 3 ijerph-17-03747-f003:**
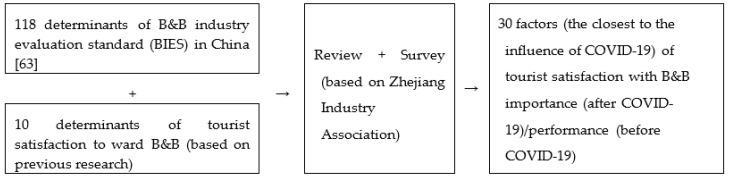
Logic of selection of the 30 questionnaire factors.

**Figure 4 ijerph-17-03747-f004:**
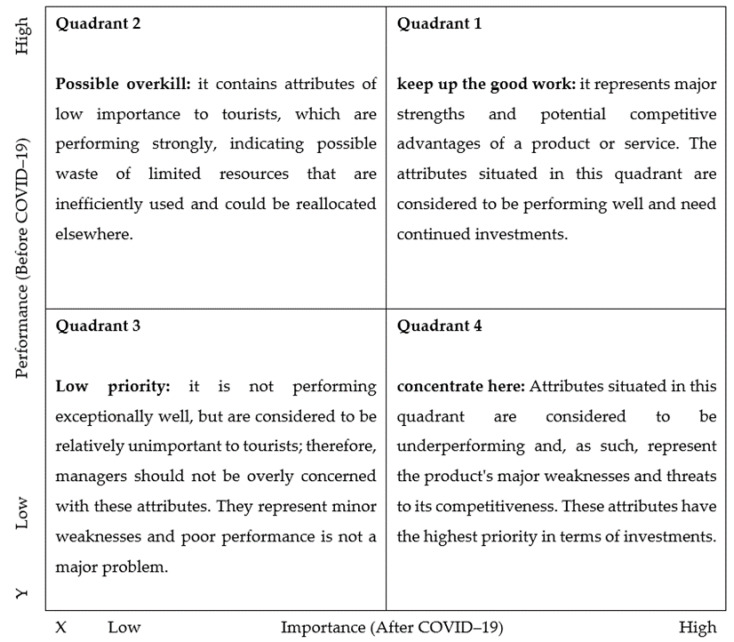
The importance–performance analysis model. Note: IA = Importance (After COVID-19), PB = Performance (Before COVID-19).

**Figure 5 ijerph-17-03747-f005:**
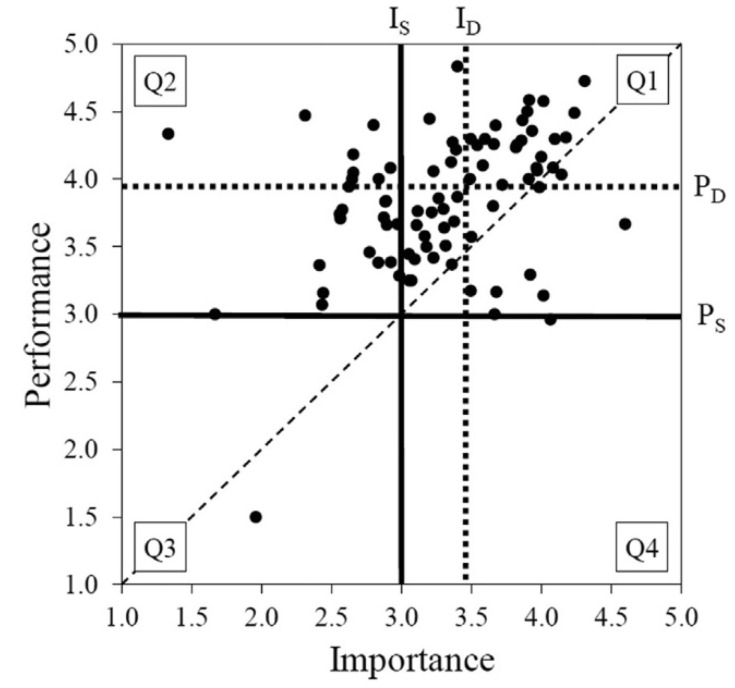
The line of different thresholds within the IPA plot. Note: Graphical comparison of scale–centered (IS − PS) and data–centered (ID − PD) approaches [22] in identifying the IPA quadrants.

**Figure 6 ijerph-17-03747-f006:**
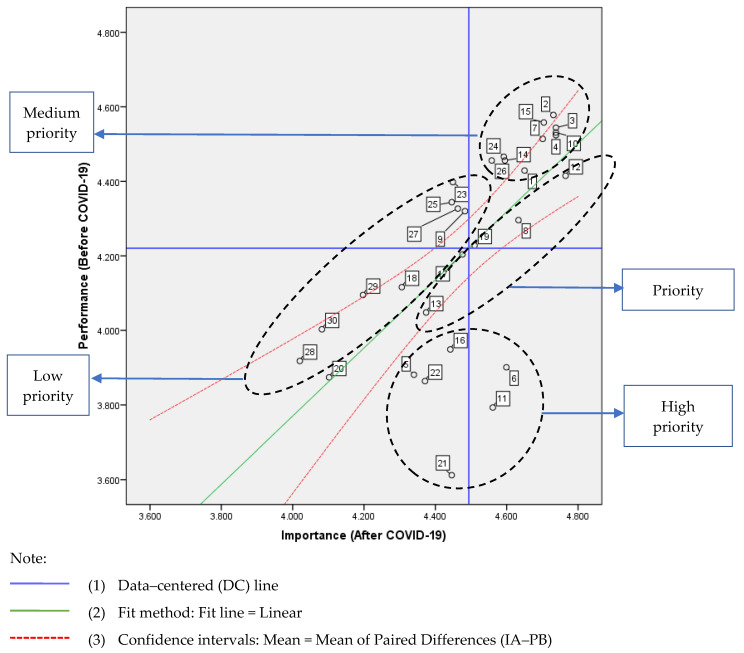
The importance (after COVID-19)-performance (before COVID-19) analysis model.

**Table 1 ijerph-17-03747-t001:** The 30 items to measure B&B experience before/after COVID-19.

Determinants of Tourist Satisfaction to Ward B&B (Based on Previous Research)	30 Factors: Importance (after COVID-19)/ Performance (before COVID-19)	QN
B&B Location [18,43,77]	Location and nearby facilities are safe and good.	1
Facility Quality [18,43,77]	The kitchen and dining room are clean and tidy.	2
	The leisure area is clean and tidy.	3
	Other service rooms are clean and tidy.	4
	Buildings are intelligent (e.g., semi–self–service management).	5
	Places or items for cleaning and disinfection are provided to tourists.	6
	The building is safe and reliable.	7
Room Quality [18,43,77]	The emergency facilities are complete (such as: first aid kit, escape equipment).	8
	The shading performance is good (e.g., opaque curtains).	9
	The rooms have plenty of natural light.	10
	Split air conditioners are used in guest rooms.	11
	Rooms are naturally ventilated.	12
	The rooms are spacious and clean.	13
	The natural landscape outside the window is good.	14
	The privacy of rooms is good.	15
Service Quality [80,81]	Contingency plans are developed and can be exercised regularly.	16
	Green consumption is encouraged and environmental protection measures are implemented.	17
Specialties [77,82]	The indoor and outdoor transition spaces are natural and beautiful (e.g., gallery frames, awnings, balconies).	18
	The outdoor space is large and natural (e.g., courtyard, terrace, roof garden).	19
	The proportion of public space is large.	20
	The layout of the rooms is scattered.	21
	Rooms of Single B&B are few and exquisite.	22
	The local culture is attractive.	23
Surrounding Environment [83]	The local people around the B&B are kind.	24
	The environment around the B&B is rural and natural.	25
Consumption Emotion [84,85]	The B & B atmosphere is good (e.g., leisurely, warm).	26
	The experience and interaction in the space is good.	27
Expectation Fulfillment [86]	B&B matches the expectations.	28
Perceived Value [87]	B&B is an important part of travel.	29
Satisfaction [88,89]	Satisfaction with the B&B.	30

Note: IA = Importance (After COVID-19), PB = Performance (Before COVID-19), QN = Question Number.

**Table 2 ijerph-17-03747-t002:** Profile of survey respondents (*n* = 588).

Variable	*n*	Percentage
Gender		
Male	260	44.22%
Female	328	55.78%
Age		
25–35	270	45.92%
36–45	122	20.75%
46–55	114	19.39%
56–65	36	6.12%
Other	46	7.82%
Educational Level		
Associate’s degree	68	11.56%
Bachelor’s degree	300	51.02%
Graduate degree	190	32.31%
Other	30	5.10%
Monthly income (US dollar)		
<500	36	6.12%
501–800	114	19.39%
801–1200	270	45.92%
1201–2000	122	20.75%
>2001	46	7.82%
Occupation		
Civil servant	20	3.40%
Company employee	202	34.35%
Student	84	14.29%
Professional	96	16.33%
Self–employed	78	13.27%
Other	108	18.37%

**Table 3 ijerph-17-03747-t003:** Validity statistics.

		Number	%
Cases	Valid	588	100
	Excludeda	0	0
	Total	588	100

**Table 4 ijerph-17-03747-t004:** Reliability Statistics.

Cronbach’s Alpha	Number of Items
0.978	30 IA + 30 PB

Note: IA = Importance (After COVID-19), PB = Performance (Before COVID-19).

**Table 5 ijerph-17-03747-t005:** Rank, means of importance and performance and paired samples (*n* = 588).

	Paired Differences (IA–PB)			IA		PB		Pearson Correlation	Sig. (2–Tailed)
QN	Mean	Rank	Std. Deviation	Mean	Rank	Mean	Rank		
21	0.833	1	1.342	4.020	30	3.918	24	0.299	0.000
11	0.769	2	1.271	4.510	15	4.228	17	0.370	0.000
6	0.697	3	1.114	4.595	11	4.456	8	0.426	0.000
22	0.507	4	1.196	4.197	27	4.095	20	0.446	0.000
16	0.493	5	1.037	4.449	19	4.398	12	0.547	0.000
5	0.459	6	1.043	4.374	23	4.048	21	0.555	0.000
12	0.350	7	0.827	4.731	5	4.578	1	0.539	0.000
8	0.337	8	0.922	4.442	22	3.949	23	0.541	0.000
13	0.327	9	0.871	4.102	28	3.874	27	0.655	0.000
19	0.282	10	0.845	4.558	14	4.456	9	0.596	0.000
17	0.272	11	0.805	4.592	12	4.466	7	0.677	0.000
20	0.228	12	1.057	4.463	18	4.327	14	0.537	0.000
1	0.221	13	0.740	4.650	8	4.429	10	0.649	0.000
4	0.214	14	0.728	4.765	1	4.415	11	0.605	0.000
10	0.207	15	0.696	4.306	26	4.116	19	0.635	0.000
3	0.194	16	0.665	4.561	13	3.793	29	0.642	0.000
18	0.190	17	0.741	4.446	20	4.344	13	0.727	0.000
7	0.187	18	0.682	4.704	6	4.558	2	0.669	0.000
9	0.163	19	0.686	4.476	17	4.204	18	0.736	0.000
2	0.153	20	0.634	4.738	3	4.531	4	0.665	0.000
15	0.146	21	0.662	4.371	24	3.864	28	0.661	0.000
14	0.139	22	0.684	4.446	21	3.612	30	0.688	0.000
27	0.136	23	0.780	4.599	10	3.901	25	0.631	0.000
24	0.126	24	0.762	4.082	29	4.003	22	0.623	0.000
28	0.102	25	0.891	4.701	7	4.514	6	0.701	0.006
29	0.102	26	0.722	4.633	9	4.296	16	0.755	0.001
25	0.102	27	0.717	4.738	4	4.524	5	0.728	0.001
26	0.102	28	0.678	4.340	25	3.881	26	0.693	0.000
30	0.078	29	0.773	4.483	16	4.320	15	0.709	0.014
23	0.051	30	0.711	4.738	2	4.544	3	0.726	0.042

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
