# Peer review of "The Impact of COVID-19 on Tourist Satisfaction with B&B in Zhejiang, China: An Importance–Performance Analysis"

_ijerph, 2020, doi:10.3390/ijerph17103747_

Round 1

Reviewer 1 Report

Thank you cery much for answering my comments. I'm glad some have been included. This time, I do not submit any other comments and congratulate the authors of the new scientific work.

Author Response

We are very grateful for your comments about the manuscript. According to your previous advice, we amended the relevant parts of the manuscript. All revisions to the manuscript have been clearly highlighted in the manuscript. After these revisions (your professional comments), the quality of this article has been greatly improved. Moreover, we have revised the problem of English and style by MDPI English editing. Thank you very much again.

Reviewer 2 Report

Thank you for resubmitting the article. I appreciate the improvements made in the article. I find the questionnaire especially useful for explaining the methodology used. The paragraphs 3.2.1. and 3.2.2. now explain IPA in sufficient depth, while the questionnaire attached provides insight into the questions asked.

It would be very useful to present the categories importance-performance in a consistent order. For example, in the Table 5, first the performance has been presented, and then the importance. For the sake of consistency, this should be reordered throughout the paper.

Also, using more readable abbreviations, e.g. importance as “Imp.” and performance as “Per.” both in text and in tables would significantly increase readability of results. I wouldn’t use IA and PB as they are not self-explanatory, but the readers need to first get familiar with the abbreviations used, The before/after notion is not important if it has been explained in the methodology and used numerous times throughout the paper in different graphs.

The article would also befit greatly from English language editing. Some lexical/grammatical notes:

Line 21: the first sentence in the abstract should be deleted. Please try to make a compelling and compact beginning to interest the readership in engaging further with the article.

Lines 25/26: please reformulate the sentence: „The time before…“ to be more clear.

Lines 28-29: please delete the sentence: „The adjusted importance….“

Line 212: you refer to the Figure 3, while it is Numbered Figure 4. Please correct this.

Lines 259-260: please correct the sentence. It begins with percentage (55,78%) and the syntax makes no sense.

Author Response

Response to Reviewer 2 Comments

We are very grateful for your comments about the manuscript. According to your advice, we amended the relevant parts of the manuscript. All revisions to the manuscript have been clearly highlighted in the manuscript. After these revisions (your professional comments), the quality of this article has been greatly improved. Moreover, thank you very much for giving us an opportunity to improve our English style. We have revised the problem of English and style by the MDPI English editing. Thank you very much again.

Point 1:

It would be very useful to present the categories importance-performance in a consistent order. For example, in the Table 5, first the performance has been presented, and then the importance. For the sake of consistency, this should be reordered throughout the paper.

Also, using more readable abbreviations, e.g. importance as “Imp.” and performance as “Per.” both in text and in tables would significantly increase readability of results. I wouldn’t use IA and PB as they are not self-explanatory, but the readers need to first get familiar with the abbreviations used, The before/after notion is not important if it has been explained in the methodology and used numerous times throughout the paper in different graphs.

Response 1:

Thank you for your suggestion on the forms and details in the manuscript. We have fixed this:

 (on line 180-183)

(1) Importance (After COVID–19): “5 = very important”, “4 = important”, “3 = so–so”, “2 = unimportant”, and “1 = very unimportant”. Appendix A shows the sample questionnaire [72][73]. (2)  Performance (Before COVID–19): “5 = very good”, “4 = good”, “3 = so–so”, “2 = not good”, and “1 = bad”.

(on line 196)

Figure 3. Logic of selection of the 30 questionnaire factors.

30 factors (the closest to the influence of COVID–19) of tourist satisfaction with B&B importance (after COVID–19)/performance (before COVID–19)

(on line 197)

Table 1. The 30 items to measure B&B experience before/after COVID–19.

30 factors: importance (after COVID–19)/ performance (before COVID–19)

(on line 289)

Table 5. Rank, means of importance, and performance and paired sample (N=588).

Paired Differences (IA–PB)

IA

 PB

Pearson Correlation

Sig. (2–tailed)

QN

Mean

Rank

Std. Deviation

Mean

Rank

Mean

Rank

21

0.833

1

1.342

4.020

30

3.918

24

0.299

0.000

11

0.769

2

1.271

4.510

15

4.228

17

0.370

0.000

6

0.697

3

1.114

4.595

11

4.456

8

0.426

0.000

22

0.507

4

1.196

4.197

27

4.095

20

0.446

0.000

16

0.493

5

1.037

4.449

19

4.398

12

0.547

0.000

5

0.459

6

1.043

4.374

23

4.048

21

0.555

0.000

12

0.350

7

0.827

4.731

5

4.578

1

0.539

0.000

8

0.337

8

0.922

4.442

22

3.949

23

0.541

0.000

13

0.327

9

0.871

4.102

28

3.874

27

0.655

0.000

19

0.282

10

0.845

4.558

14

4.456

9

0.596

0.000

17

0.272

11

0.805

4.592

12

4.466

7

0.677

0.000

20

0.228

12

1.057

4.463

18

4.327

14

0.537

0.000

1

0.221

13

0.740

4.650

8

4.429

10

0.649

0.000

4

0.214

14

0.728

4.765

1

4.415

11

0.605

0.000

10

0.207

15

0.696

4.306

26

4.116

19

0.635

0.000

3

0.194

16

0.665

4.561

13

3.793

29

0.642

0.000

18

0.190

17

0.741

4.446

20

4.344

13

0.727

0.000

7

0.187

18

0.682

4.704

6

4.558

2

0.669

0.000

9

0.163

19

0.686

4.476

17

4.204

18

0.736

0.000

2

0.153

20

0.634

4.738

3

4.531

4

0.665

0.000

15

0.146

21

0.662

4.371

24

3.864

28

0.661

0.000

14

0.139

22

0.684

4.446

21

3.612

30

0.688

0.000

27

0.136

23

0.780

4.599

10

3.901

25

0.631

0.000

24

0.126

24

0.762

4.082

29

4.003

22

0.623

0.000

28

0.102

25

0.891

4.701

7

4.514

6

0.701

0.006

29

0.102

26

0.722

4.633

9

4.296

16

0.755

0.001

25

0.102

27

0.717

4.738

4

4.524

5

0.728

0.001

26

0.102

28

0.678

4.340

25

3.881

26

0.693

0.000

30

0.078

29

0.773

4.483

16

4.320

15

0.709

0.014

23

0.051

30

0.711

4.738

2

4.544

3

0.726

0.042

(on line 290)

4.2. Importance (after COVID–19)–Performance (before COVID–19) Analysis (IPA)

(on line 388)

Appendix A

The sample questionnaire

All abbreviations of this manuscript are explained in the introduction (limited space in many tables)(on line 85):

Figure 1. The logical model.

Notes: B&B = Bed and Breakfast, BIES = B & B industry evaluation standard, COVID–19 = Corona Virus Disease 2019, IPA = importance–performance analysis, IA = Importance (After COVID–19), PB = Performance (Before COVID–19), QN = Question Number, TS = tourist satisfaction

Point 2:

Line 21: the first sentence in the abstract should be deleted. Please try to make a compelling and compact beginning to interest the readership in engaging further with the article.

Lines 25/26: please reformulate the sentence: „The time before…“ to be more clear.

Lines 28-29: please delete the sentence: „The adjusted importance….“

 Response 2:

We are very grateful for your comments about the details in the abstract. We have revised this problem:

(on line 21-31)

After the outbreak of COVID-19 (especially in the stage of tourism recovery), the bed and breakfast (B&B) tourism industry faced big challenges in improving its health strategies. B&Bs are very important for the tourism industry in China and many other countries. However, few studies have studied the impact of B&Bs, under COVID–19, on tourism in China. Our paper is among one of the first studies to investigate the impact of COVID–19 on tourist satisfaction with B&Bs in China. The work/travel restrictions started from 20 January 2020, and work/after travel resumed from 20 February 2020 in Zhejiang, China. Data were collected from 588 tourists (who experienced B&Bs in Zhejiang, China) from a WeChat online survey, from 1 March to 15 March 2020. The current study attempted to fill the gap by studying the changing tourist satisfaction levels with B&Bs before/after COVID–19. Moreover, some suggestions are given to the B&B industry for tourism resumption after COVID–19 by an importance–performance analysis (IPA).

Point 3:

Line 212: you refer to the Figure 3, while it is Numbered Figure 4. Please correct this.

Response 3:

Thank you for your suggestion on the details in the manuscript. We have fixed this:

(on line 213)

Figure 4 shows the IPA model.

Point 4:

Lines 259-260: please correct the sentence. It begins with percentage (55,78%) and the syntax makes no sense.

Response 4: We are very grateful for your comments about the details. We have revised this problem:

(on line 261).

Among the 588 tourists, 55.78% were women and 44.22% were men.

This manuscript is a resubmission of an earlier submission. The following is a list of the peer review reports and author responses from that submission.

Round 1

Reviewer 1 Report

Many thanks to the authors for this important article. In my opinion, this article is well prepared and suitable for publication after minor corrections, which, however, are rather suggestive.
If I understood correctly, the article was finished writing on April 1, 2020, so I propose to update the data that was described in the introduction. I suggest that the authors reflect on specifying the examined issue, because they examined customer satisfaction in the period 1-15 March 2020 and not after the end of the pandemic (maybe a good idea is to determine: local cessation of the pandemic). My remark concerns the fact that the pandemic on other continents (from which respondents may have come) began after March 15 (e.g. in my country), and during the research period the borders were still open. Do the authors know what nationality the respondents were and how delaying eopidemia could affect their research? The situation after the opening of borders by European and American countries may be interesting. The impact of COVID-19 on tourism will certainly be much more severe than in the case of SARS.
I consider the selection of literature to be appropriate. Possibly some shortcomings in this area are justified by the fact that this is an innovative topic.
I really like the concept of the four-field matrix (although it is a bit similar to BCG). Maybe it is worth thinking about the quantification of individual groups of factors in the form of synthetic variables?
I have doubts about the significance level of individual factors in Table 2 and row 328, because this significance is probably not 0.000, but rather <0.001.
I thank the authors for interesting information and wish them further scientific successes.

Author Response

Response to Reviewer 1 Comments

We are very grateful for your comments about the manuscript. According to your advice, we amended the relevant parts of the manuscript. All revisions to the manuscript have been clearly highlighted in the manuscript. After these revisions (your professional comments), the quality of this article has been greatly improved. Thank you very much again.

Point 1: If I understood correctly, the article was finished writing on April 1, 2020, so I propose to update the data that was described in the introduction.

Response 1: Thank you for your suggestion on the data updating in the manuscript. The new data that was described in the introduction “As of May 1, 2020, 3,175,207 cases have been confirmed globally. It is the latest infectious disease to rapidly develop worldwide” (on line X39).

Point 2: I suggest that the authors reflect on specifying the examined issue, because they examined customer satisfaction in the period 1-15 March 2020 and not after the end of the pandemic (maybe a good idea is to determine: local cessation of the pandemic). My remark concerns the fact that the pandemic on other continents (from which respondents may have come) began after March 15 (e.g. in my country), and during the research period the borders were still open. Do the authors know what nationality the respondents were and how the delaying epidemic could affect their research? The situation after the opening of borders by European and American countries may be interesting. The impact of COVID-19 on tourism will certainly be much more severe than in the case of SARS.

Response 2: We are very grateful for your comments about the time and areas of the questionnaires. There were some reasons for this choice.

First, due to “Zhejiang has resumed work gradually from February 20, 2020.” (on line 56) and “The time of before/after satisfaction was before the work/travel restrictions (from 20 January 2020) to work/after travel resumption (after 20 February 2020). Data was collected from 588 tourists of B&B in Zhejiang, China, lasting from 1 March to 15 March 2020. The adjusted importance (before COVID–19)–performance (after COVID–19) analysis (IPA) was used.” (on line 65-68), so “There were 588 responses that were selected for the analysis. and the responses were collected from 1 March to 15 March 2020.” (on line 74-75).

Second, “There were 588 responses who have experienced B&B in Zhejiang before the COVID–19 that were selected for the analysis.” (on line 170-172), so we only choose the tourists in Zhejiang, China.

Third, as your brilliant ideas of “My remark concerns the fact that the pandemic on other continents (from which respondents may have come) began after March 15 (e.g. in my country), and during the research period the borders were still open. The situation after the opening of borders by European and American countries may be interesting.”, we suggest that researchers in other parts of China and on other continents work together to produce similar studies, thereby creating a worldwide body of literature examining the phenomena related to the effects of certain types of crisis impact on B&B and tourism. Hence, future studies are necessary to use a greater sampling range. Future studies should also incorporate more nuanced research questions.

Point 3: I really like the concept of the four-field matrix (although it is a bit similar to BCG). Maybe it is worth thinking about the quantification of individual groups of factors in the form of synthetic variables?

Response 3: Thank you for your suggestion on the concept of the four-field matrix in the manuscript. It is worth thinking about the quantification of individual groups of factors in the form of synthetic variables. However, because the word count of this article needs to be controlled, we will accept your perfect suggestions for the next article.

Point 4: I have doubts about the significance level of individual factors in Table 2 and row 328, because this significance is probably not 0.000, but rather <0.001.

Response 4: We are very grateful for your comments about the details. We have revised this problem. “Most of the importance and performance means were found to be significantly different (Sig. 2–tailed) at the <0.01 level (QN. 23/25/28/29/30 <0.05).” (on line 287-289).

Point 5: I thank the authors for interesting information and wish them further scientific successes.

Response 5: We are very grateful for your comments about this manuscript again. After these revisions (your professional comments), the quality of this article has been greatly improved. Thank you very much again.

Reviewer 2 Report

Line 21: please correct at the end of the line- “&B”.

Lines 37 to 41: please rewrite the UPPER CASE parts and write them in a regular lower case style, as the rest of the text.

Line 47-53: please use lower case first letter after the semicolon. This is a correct way of writing: …of SARS in 2003; at the time….

Line 57: please put space after the closed bracket. Correct way of writing is: ..areas (e.g. B&B in the countryside)

Throughout the text: please put spaces after the square bracket. Correct way of writing is: ….to modify tourism development [8].

Lines 109-110: please explain due to what event in 2003. have the tourism arrivals dropped by 1.2%. I understand what the authors are trying to say, but it needs to be clearly stated here.

Figure 2.: why does it state in the graph “SARA”? Shouldn’t it be SARS, as it is in the caption and in the text?

Line 215: please use same font size as the rest of the text

Line 226 and onwards: the 30 items measured are just fine, but I do not agree with the parallel use of both IPA and the t-test. If the variables measured are different (importance vs. performance), than no t-test can be deployed, because the difference between these two concepts need not be tested, as they are proven in the literature to be different. However, if the question was what is tourists satisfaction before COVID 19 and after, than IPA cannot be used and t-test makes a lot of sense. If the questions have really been posed the way presented here, than it cannot be analyzed with either IPA ot t-test, as it is completely inconsistent:

Lines 201-202: (1) Performance (After COVID–19): “5 = very good ”, “4 = good”, “3 = so–so”, “2 = not good”, and “1 201 = bad”. (2) Importance (After COVID–19): “5 = very important”, “4 = important”, “3 = so–so”, “2 = 202 unimportant”, and “1 = very unimportant”.

Table 5: IA = Importance (After COVID–19), PB = Performance (Before COVID–19)

Line 332: Iimportance (before COVID–19)Performance (after COVID–19)

The above-mentioned inconsistency leads to the question whether the actual question posed to the tourists has ben truthfully presented here? The authors should start by consistently and truthfully presenting what has been an actual question posed to the tourists, and than decide on the best way to interpret the results. The zig zag with importance and performance + before COVID19 and after COVID 19 has no logical basis.

Author Response

Response to Reviewer 2 Comments

We are very grateful for your comments about the manuscript. According to your advice, we amended the relevant parts of the manuscript. All revisions to the manuscript have been clearly highlighted in the manuscript. After these revisions (your professional comments), the quality of this article has been greatly improved. Moreover, thank you very much for giving us an opportunity to improve our English style. We have revised the problem of English and style. We hope that our manuscript will be easier to understand. After the revision, if needed, we will choose the English editing from MDPI to improve the English. Thank you very much again.

Point 1: Line 21: please correct at the end of the line- “&B”.

Response 1: Thank you for your suggestion on the form in the manuscript. We have fixed this: “B&B was very important for the tourism industry in China and many other countries.” (on line 21).

Point 2: Lines 37 to 41: please rewrite the UPPER CASE parts and write them in a regular lower case style, as the rest of the text.

Response 2: We are very grateful for your comments about the details. We have revised this problem: “Corona virus disease 2019 (COVID–19) is a highly infectious disease with a long incubation period [1]. Due date May 1, 2020, 3,175,207 cases have been confirmed globally [2]. It was the latest infectious disease to rapidly develop worldwide [3]. 27 cases of the unknown virus were reported on 31st December 2019 [4]. An estimated 60 million residents of Wuhan and many other cities in China were subjected to community containment measures from 23 January 2020. These large–scale types of actions have never been used in the past (even the SARS in China) [5].” (on line 36-41).

Point 3: Line 47-53: please use lower case first letter after the semicolon. This is a correct way of writing: …of SARS in 2003; at the time….

Response 3: Thank you for your suggestion on the form in the manuscript. We have fixed this: “China, as the world's most populous nation and the world's second-largest economy, had already battled with the epidemic (the SARS);at the time, however, it was 4% of the global total – it was now 17%.” (on line 43-45).

Point 4: Line 57: please put space after the closed bracket. Correct way of writing is: …areas (e.g. B&B in the countryside)

Response 4: We are very grateful for your comments about the details. We have revised this problem: “Nature-based areas (e.g. B&B in countryside) were likely to be the target destinations [10]. New motivations to travel to nature–based areas became the evident with SARS [11]. A potential marketing emphasis that nature–based tourism types (e.g. nature–based B&B) could be invigorated and expanded after the COVID–19 [12].” (on line 49-52).

Point 5: Throughout the text: please put spaces after the square bracket. Correct way of writing is: ….to modify tourism development [8].

Response 5: Thank you for your suggestion on the form in the manuscript. We have fixed this: “In 2003, a window of opportunity to modify tourism development was opened by the crisis of SARS [8].” (on line 48).

Point 6: Lines 109-110: please explain due to what event in 2003. have the tourism arrivals dropped by 1.2%. I understand what the authors are trying to say, but it needs to be clearly stated here.

Response 6: Thank you for your suggestion. Explanation has been added: “According to world tourism organization (WTO), in 2003, tourism arrivals felled 1.2% to 694 million (compared to the same period in 2002) in China, and hotel occupancy rates felled down to 10%.” (on line 92=94).

Point 7: Figure 2.: why does it state in the graph “SARA”? Shouldn’t it be SARS, as it is in the caption and in the text?

Response 7: We are very grateful for your comments about the details. We are sorry for the spelling mistakes. We have revised this problem in figure 2. Thank you very much again.

Point 8: Line 215: please use same font size as the rest of the text

Response 8: We are very grateful for your comments about the details. Due to some typographical errors, the size (on line 215) was not the same font size as the rest of the text. We have revised this problem in the text: “Ten determinants of tourist satisfaction were identified [60][35]. Based on previous researches and B & B industry evaluation standard (BIES) in China (Table 1, Figure 3), factors were generated.” (on line 189-190).

Point 9: Line 226 and onwards: the 30 items measured are just fine, but I do not agree with the parallel use of both IPA and the t-test. If the variables measured are different (importance vs. performance), then no t-test can be deployed, because the difference between these two concepts need not be tested, as they are proven in the literature to be different. However, if the question was what is tourist satisfaction before COVID 19 and after, then IPA cannot be used and t-test makes a lot of sense. If the questions have really been posed the way presented here, then it cannot be analyzed with either IPA or t-test, as it is completely inconsistent:

Response 9: We are very grateful for your comments about the correlations between IPA and t-test. We are sorry for the mistakes. We have revised this problem in Table 5. We have deleted the t-test: “Most of the importance and performance means were found to be significantly different (Sig. 2–tailed) at the <0.01 level (QN. 23/25/28/29/30 <0.05). Variables in each category are ranked in order by Paired Differences (IA–PB).” (on line 287-290).

Table 5. Rank, means of importance, and performance and paired sample (N=588).

Paired Differences (IA–PB)

PB

IA

Pearson Correlation

Sig. (2–tailed)

QN

Mean

Rank

Std. Deviation

Mean

Rank

Mean

Rank

21

0.833

1

1.342

3.918

24

4.020

30

0.299

0.000

11

0.769

2

1.271

4.228

17

4.510

15

0.370

0.000

6

0.697

3

1.114

4.456

8

4.595

11

0.426

0.000

22

0.507

4

1.196

4.095

20

4.197

27

0.446

0.000

16

0.493

5

1.037

4.398

12

4.449

19

0.547

0.000

5

0.459

6

1.043

4.048

21

4.374

23

0.555

0.000

12

0.350

7

0.827

4.578

1

4.731

5

0.539

0.000

8

0.337

8

0.922

3.949

23

4.442

22

0.541

0.000

13

0.327

9

0.871

3.874

27

4.102

28

0.655

0.000

19

0.282

10

0.845

4.456

9

4.558

14

0.596

0.000

17

0.272

11

0.805

4.466

7

4.592

12

0.677

0.000

20

0.228

12

1.057

4.327

14

4.463

18

0.537

0.000

1

0.221

13

0.740

4.429

10

4.650

8

0.649

0.000

4

0.214

14

0.728

4.415

11

4.765

1

0.605

0.000

10

0.207

15

0.696

4.116

19

4.306

26

0.635

0.000

3

0.194

16

0.665

3.793

29

4.561

13

0.642

0.000

18

0.190

17

0.741

4.344

13

4.446

20

0.727

0.000

7

0.187

18

0.682

4.558

2

4.704

6

0.669

0.000

9

0.163

19

0.686

4.204

18

4.476

17

0.736

0.000

2

0.153

20

0.634

4.531

4

4.738

3

0.665

0.000

15

0.146

21

0.662

3.864

28

4.371

24

0.661

0.000

14

0.139

22

0.684

3.612

30

4.446

21

0.688

0.000

27

0.136

23

0.780

3.901

25

4.599

10

0.631

0.000

24

0.126

24

0.762

4.003

22

4.082

29

0.623

0.000

28

0.102

25

0.891

4.514

6

4.701

7

0.701

0.006

29

0.102

26

0.722

4.296

16

4.633

9

0.755

0.001

25

0.102

27

0.717

4.524

5

4.738

4

0.728

0.001

26

0.102

28

0.678

3.881

26

4.340

25

0.693

0.000

30

0.078

29

0.773

4.320

15

4.483

16

0.709

0.014

23

0.051

30

0.711

4.544

3

4.738

2

0.726

0.042

Point 10:

  • Lines 201-202: (1) Performance (After COVID–19): “5 = very good”, “4 = good”, “3 = so–so”, “2 = not good”, and “1 201 = bad”. (2) Importance (After COVID–19): “5 = very important”, “4 = important”, “3 = so–so”, “2 = 202 unimportant”, and “1 = very unimportant”.
  • Table 5: IA = Importance (After COVID–19), PB = Performance (Before COVID–19)
  • Line 332: Importance (before COVID–19)–Performance (after COVID–19)
  • The above-mentioned inconsistency leads to the question of whether the actual question posed to the tourists has been truthfully presented here? The authors should start by consistently and truthfully presenting what has been an actual question posed to the tourists, and then decide on the best way to interpret the results. The zig-zag with importance and performance + before COVID19 and after COVID 19 has no logical basis.

Response 10: This point has helped us a lot. It pointed out the main problem in the logical basis of our manuscript. Because we did not carefully check the position of the “before/after” in the brackets, which caused these logical problems. Your comments also helped us improve the quality of the article. We have modified the expression of these parts:

  • “The adjusted importance (after COVID–19)–performance (before COVID–19) analysis (IPA) was a new attempt. The current study attempted to fill the gap by studying the changing of tourist satisfaction with B&B before/after COVID–19. Moreover, some suggestions would be given to the B&B industry for tourism resumption after the COVID–19 by the importance–performance analysis (IPA).” (on line 27-31).
  • “The adjusted importance (before COVID–19)–performance (after COVID–19) analysis (IPA) was used. The current study attempted to fill the research gap by investigating the changing of tourist satisfaction with B&B before/after COVID–19. Moreover, some suggestions would be given to the B&B industry to recover after the COVID–19 by the importance–performance analysis (IPA).” (on line 67-71).
  • “(1) Performance (Before COVID–19): “5 = very good”, “4 = good”, “3 = so–so”, “2 = not good”, and “1 = bad”. (2) Importance (After COVID–19): “5 = very important”, “4 = important”, “3 = so–so”, “2 = unimportant”, and “1 = very unimportant”.” (on line 176-178).
  • “The adjusted importance (after COVID–19)–performance (before COVID–19) analysis (IPA) was a new attempt.” (on line 362-363).
  • “Note: IA = Importance (After COVID–19), PB = Performance (Before COVID–19), QN = Question Number” (on Table 5).
  • “Figure 6. The importance (After COVID–19)–performance (Before COVID–19) analysis model.” (in Figure 6).

Reviewer 3 Report

I recommend not to divide the introduction into sections.

In the conclusions, the theoretical and practical implications should be clearly included.

I recommend to move Limitations and Future Research in the Conclusions.

Author Response

Response to Reviewer 3 Comments

We are very grateful for your comments about the manuscript. According to your advice, we amended the relevant parts of the manuscript. All revisions to the manuscript have been clearly highlighted in the manuscript. After these revisions (your professional comments), the quality of this article has been greatly improved. Thank you very much again.

Point 1: I recommend not to divide the introduction into sections.

Response 1: Thank you for your suggestion on the form in the manuscript. We have revised the introduction: “Corona virus disease 2019 (COVID–19) is a highly infectious disease with a long incubation period [1]. Due date May 1, 2020, 3,175,207 cases have been confirmed globally [2]. It was the latest infectious disease to rapidly develop worldwide [3]. 27 cases of unknown virus were reported on 31st December 2019 [4]. An estimated 60 million residents of Wuhan and many other cities in China were subjected to community containment measures from 23 January 2020. These large–scale types of actions have never been used in the past (even the SARS in China) [5]. …This article contained six sections. Section 1 was the introduction. Section 2 contained the literature review. Section 3 was data collection and research methods. Section4 was the results. Section 5 comprised the impacts and limitations. The 6 section was the conclusions.” (on line 36-87).

Point 2: In the conclusions, the theoretical and practical implications should be clearly included. I recommend to move Limitations and Future Research in the Conclusions.

Response 2: This point has helped us a lot. It pointed out the main problem in the conclusion part of our manuscript. Your comments also helped us improve the quality of the article. We have modified the expression of this part:

“B&B is very important for the tourism industry in many countries, and it is especially welcomed by tourists in recent years in China. To our knowledge, our study was among one of the first studies to investigate the immediate impact of the COVID–19 pandemic on tourist satisfaction with B&B in China. Many previous studies have reported the COVID–19. Some others studied the correlations of the COVID–19 and quality of life in China. But few studies have reported the impact of B&B under COVID–19 on tourism in China. The adjusted importance (after COVID–19)–performance (before COVID–19) analysis (IPA) was a new attempt. Moreover, some promotion suggestions were given to the B&B industry recovery after the COVID–19 by the IPA.”

“On the other hand, there were some limitations and future research. First, the data was collected from tourists of B&B in Zhejiang, China. So, it was somewhat difficult to apply the suggestions of the impact of the COVID–19 to other areas. Future researches may expand this scope. Second, although we identified the relationships between the determinants of tourist satisfaction and the COVID–19, the relative strength of these correlations was unknown. We can test the model and identify the degree of influence about the correlations between these factors to promote the B&B industry in further research. More nuanced research questions should be incorporated. Third, the current paper employed an IPA approach. Even though this method was a widely known method in the tourism industry, it was also a new attempt to the B&B. Thus, we suggest that researchers in other parts of China and on other continents work together to produce similar studies, thereby creating a worldwide body of literature examining the phenomena related to the effects of crisis (e.g. COVID–19) impact on B&B and tourism.” (on line 357-376).

Round 2

Reviewer 2 Report

I appreciate the time the authors have taken to amend and extend the manuscript. The items development is now clearer. However, the logic behind the questionnaire still needs to be considerably enhanced. You should answer the following questions regarding the questionnaire in the methodology section, which also needs to be reflected in the introduction and abstract: Whom did you ask? Where did you ask? How did you ask? What did you ask? Why did you ask precisely that and not something else?

  1. Please provide the questionnaire in the Annex, as this is a novel methodology and researchers need to be able to replicate the study. It can also help reviewers a lot in understanding the objectives and methods used. Example for this can be found on Google Scholar: Dressler, M. (2017). Strategic profiling and the value of wine & tourism initiatives. International Journal of Wine Business Research, 29(4), 484-502. doi:10.1108/ijwbr-04-2017-0026
  2. Lines 170.-172.: Please provide a clearer explanation as to how and where has the questionnaire been distributed: online, in the hotel reception, in the mall, on the street or some other setting? What about the relevant sample characteristics: hotel customers, attraction visitors, nationality? What was the criterion to include/exclude the answers, having in mind the inter-temporal nature of the questions? A visit to any hotel in Zhejiang both before and after COVID-19? A visit to any hotel in China both before and after COVID-19? Did they rate actual encounters or imagined ones?
  3. Lines 173.-180.: Did you have three scales (three questions) for each of the 30 items? What was the precise question asked for each of the 3 questions? The current presentation doesn’t answer to many questions:

performance (before COVID-19) (performance of what for whom, where and precisely when?)

importance (after COVID-19) (importance of what for whom, where and precisely when?)

Tourists’ perception of B&B during their stay. (satisfaction level of tourists or of hotel guests?)

This would be examples of questions you have possibly asked your respondents:

Please rate the performance of a certain aspect of hotel service during your stay in hotel xy before COVID-19?

5 = very good”, “4 = good”, “3 = so–so”, “2 = not good”, and “1 = bad

Please rate the importance of a certain aspect of hotel service for your present stay in hotel xy or future stay in any hotel (after COVID -19)?

5= very important”, “4 = important”, “3 = so–so”, “2 = unimportant”, and “1 = very unimportant

Please rate your current stay? (this would imply that the sample consists of hotel guests, but it hasn’t been explicitly stated anywhere in the text)

highly satisfactory, “satisfactory”, “so–so”, “unsatisfactory” and 179 “highly unsatisfactory

Line 186: please delete: “Content.”, in the middle of the row

Line 190: please delete “T” at the end of the row

Lines 110-121 and 149-164: Destination image and tourist satisfaction are also an important tool used by DMOs to actively research and manage the perceptions of tourists about the destination. See: Paunovic, I. (2014). Satisfaction of tourists in Serbia, destination image, loyalty and DMO service quality. European Journal of Tourism, Hospitality and Recreation, 3(1), 163-181.

Reviewer 3 Report

I suggest that we further enhance the conclusions.
As things stand at present, they still appear to be rather weak.